# Risk-Perception Change Associated with COVID-19 Vaccine’s Side Effects: The Role of Individual Differences

**DOI:** 10.3390/ijerph19031189

**Published:** 2022-01-21

**Authors:** Laura Colautti, Alice Cancer, Sara Magenes, Alessandro Antonietti, Paola Iannello

**Affiliations:** 1Department of Psychology, Catholic University of the Sacred Heart, 20123 Milan, Italy; alice.cancer@unicatt.it (A.C.); sara.magenes@unicatt.it (S.M.); alessandro.antonietti@unicatt.it (A.A.); paola.iannello@unicatt.it (P.I.); 2Fraternità e Amicizia Società Cooperativa Sociale ONLUS, 20146 Milan, Italy

**Keywords:** COVID-19, vaccine, risk-perception, individual differences, conspiracy theories, analytic thinking, decision making

## Abstract

The COVID-19 vaccine appears to be a crucial requirement to fight the pandemic. However, a part of the population possesses negative attitudes towards the vaccine. The spread of conspiracy theories and contradictory information about the pandemic have altered the population’s perception of risk. The risk-perception of the vaccine’s side effects may be affected by individual differences. The complex relationship between risk-perception and individual differences is relevant when people have to make decisions based on ambiguous and constantly changing information, as in the early phases of the Italian vaccination campaign. The present study aimed at measuring the effect of individual differences in risk-perception associated with the COVID-19 vaccine’s side effects in a context characterized by information ambiguity. An online survey was conducted to classify a sample of Italian pro-vaccine people into cognitive/behavioral style groups. Furthermore, changes in vaccine risk-perception after inconsistent communications regarding the vaccine’s side effects were compared between groups. The results showed that “analytical” individuals did not change their perception regarding the probability of vaccine side effects but changed their perception regarding the severity of side effects; “open” and “polarized” individuals neither changed their perception regarding the probability nor of the severity of side effects, showing a different kind of information processing, which could interfere with an informed decision-making process.

## 1. Introduction

More than 20 months after the announcement of the COVID-19 spread, we are still in the midst of a worldwide unprecedented emergency, from different points of view, including health, social, and economic perspectives [1]. Despite the restrictive measures adopted by the whole world, the vaccine seems to be the only way to get back to a “normal” life. Since the early stage of the vaccination campaign, which began in December 2020 in most European countries (including Italy), serious side effects have been recorded for all types of vaccines. In detail, Italy’s eighth COVID-19 Vaccine Pharmacovigilance Report [2] described side effects in 101,110 cases out of 84,010,605 doses administered (reporting rate of 120 per 100,000 doses), 14.4% of which consisted of severe events, with a rate of 21 serious events per 100,000 doses. Specifically, the deaths from the beginning of the vaccination campaign reached 608 per 100,000 doses. Meanwhile, 85.4% reported non-severe events, such as pain at the injection site, fever, asthenia/fatigue, and muscle aches.

Globally it is assumed that the benefits of vaccines turn out to be greater than the risks, as indicated by authorities, such as the WHO and the EMA [1]. Despite this, especially during the first months of the Italian vaccination campaign, controversial decisions about the inoculation of AstraZeneca doses occurred and the spread of conflicting and inconsistent information regarding the safety of vaccines was circulated by the mass media. In particular, on the 30 January 2021 AIFA (the Italian Medicines Agency) firstly recommended the inoculation of AstraZeneca to people between 18 and 55 years old. On the 15 February, the administration of this vaccine was limited to subjects between 55 and 65 years of age. After the occurrence of rare side effects, such as unusual blood clots with low blood platelets, on the 15 March 2021 some European countries, included Italy, suspended as a precautionary measure AstraZeneca inoculations. Later, following in-depth investigations about the vaccine’s safety, on the 19 March 2021 the administration of this vaccine was started again and from the 3 June 2021 it was recommended only for people over 65 years old. Finally, the contract for AstraZeneca’s supplies has not been renewed for 2022 [2]. Consequently, increasing mistrust and hesitancy towards vaccines has occurred in the Italian population.

Vaccine hesitancy, a delay in acceptance, or a refusal of vaccination despite availability of vaccination services has been identified as one of the greatest threats to public health at a global level [3]. A refusal rate of more than 10% is estimated to be sufficient to undermine the population benefits of vaccination against COVID-19 [4]. Numerous studies have shown that a part of the population shares negative attitudes towards the COVID-19 vaccine. Some of the reasons are attributable to the hasty development of the vaccine [5], a diminished trust in research and vaccination [6], and widespread misinformation about the pandemic and its management [7].

Accordingly, the present article aims at exploring the individual characteristics and the risk perception change of those who are pro-vaccine within a group of Italian people.

### 1.1. Conspiracy Theories

The spread of conspiracy theories and fake news about the pandemic and health conditions have altered the population’s perception of risk, by inducing those who support similar ideas to engage in behavior contrary to public welfare, including a negative attitude towards the vaccine [8]. Conspiracy theories have been defined as attempts to explain significant social and political events through claims of secret plots orchestrated by powerful and malevolent actors [9]. Most theories include the belief that: (a) COVID-19 is a man-made virus created to control the population, possibly via 5G technology; (b) COVID-19 is produced to ruin the economy; and (c) vaccines induce abnormal side effects in the entire population [10,11,12]. As a consequence, the belief in conspiracy theories which affect levels of trust regarding the effectiveness and safety of the COVID-19 vaccines [3,6,13], have remained unchanged throughout the pandemic [8]. Consequently, in the context of the current pandemic, the belief in COVID-19 conspiracy theories is found to decrease institutional trust [6,14], support for government regulations, and the use of preventive measures (i.e., wearing a face mask, maintaining social distance, and frequent handwashing) [15,16,17]. Studies have found that negative attitudes towards experts make people more likely to deny scientific consensus and endorse conspiracies that oppose it [14,18,19,20]. Knowledge about such an issue and the educational level possessed by an individual do not affect the adherence to conspiratorial thinking (i.e., a person highly knowledgeable about politics, but with low trust in the institutions, is very prone to conspiracies [21]). On the other hand, trust in research and science plays a crucial role in adopting preventive behaviors during the COVID-19 pandemic and developing a positive attitude towards vaccines [22,23].

### 1.2. Cognitive Style

An individual characteristic that could either support or undermine trust in vaccines is an individual’s personal cognitive style. According to the distinction between analytic and impulsive styles, analytic thinking—which is slow and deliberative—requires a willingness to evaluate empirical evidence and to weigh the available data, making choices less susceptible to biases [24,25]. Moreover, it is positively related to general health beliefs, to vaccine attitude [26], and to the trust in authorities and science [27,28]. On the other hand, impulsivity conveys unplanned, careless, and gut-level behaviors and it is characterized by poor inhibitory response control and fast information processing [29]. Regarding this, several studies showed that impulsive thinking is related to dangerous behaviors which lead to negative consequences for health [30]. Furthermore, impulsivity was found to negatively correlate to the adoption of preventive measures to counteract the spread of COVID-19 [31], to predict the embracing of prescriptions based on alternative medicines, and can result in the development of low vaccination adherence [32]. In addition to this, it is positively correlated to the tendency to believe in fake news and to endorse conspiracy theories related to COVID-19 (see for example [31,33,34]).

Another individual difference is the ability to tolerate ambiguous situations (e.g., situations that “cannot be adequately structured or categorized by an individual because of the lack of sufficient cues” [35] (p. 30)). It might impact the way people perceive, experience, and interpret events. Specifically, when faced with unclear and unstructured situations, individuals with low tolerance for ambiguity perceive them as a source of discomfort and threat [36,37,38], which is often associated with the experience of a lack of perceived control over the situation [39]. 

Like people with a high intolerance to uncertainty, individuals with high need for cognitive closure (NCC) may make poor decisions when situations are uncertain [40,41,42]. Thus, individuals with high NCC tend to quickly catch information in order to reduce ambiguity and to maintain acquired knowledge [40]. Thus, to face the need for closure, individuals high in NCC often use heuristics and simple information that is readily available to obtain fast and certain answers towards uncertain situations [42]. The literature indicates that individual differences in NCC can be linked to mental distress and higher levels of anxiety during unpredictable situations [41,42,43], factors that lead to looking for solutions to “close” and reduce uncertainty. 

The existence of a relationship between intolerance for ambiguity, NCC, and risk-taking propensity was reported in several studies (i.e., [44,45,46]). Furthermore, individuals’ reactions to risk and ambiguity tend to be consistent over time [47]. 

### 1.3. Risk-Perception

The perception of risk can be defined as an individual difference in the way of processing information and reacting to risky events [48]. It is crucial to health behavior and it is fundamental for making decisions concerning the reduction of specific health threats [49]. It is assumed that risk-perception is composed of two dimensions, which concern the probability and the severity of the consequences related to an event [50,51].

Slovic and Peters [52] stated that people process risk through two routes: “risk as feelings”, which is related to our visceral and automatic responses to danger, and “risk as analysis”, which refers to logical reasoning and scientific deliberation to manage the decisional process and the assessment of risk. Studies showed that when people judge or make decisions regarding a risky situation, they rely more on the emotional and affective component of the event than on objective data [53,54]. Thus, affect represents a cue to make judgments and it can bias the perception of the probability of occurrence of an event. In fact, the perception of risk or of negative consequences of sporadic and sensational events may be overestimated if they are strongly emotional connoted, due to the higher possibility to recall them in one’s own environment (the so-called availability heuristic [55]). In this way, media can play an important role in emotionally connoting dramatic events, including the vaccine’s side effects, making them more “available”.

Moreover, it is plausible that risk-perception of the vaccine’s side effects can be affected by the individual characteristics previously described, i.e., a tendency to believe in conspiracy theories, cognitive style, impulsivity, tolerance for ambiguity, and desire for cognitive closure. The complex relationships between these factors appear to be particularly relevant when people are asked to make decisions based on ambiguous and constantly changing information, such as in the early phases of the vaccination campaign in Italy.

## 2. Materials and Methods

### 2.1. Aims

The present study aimed at measuring the effect of individual differences in the risk-perception change following the reports of side-effects of the vaccine in a context characterized by information ambiguity. More precisely, variation in risk-perception was investigated in the period following the reports of rare side-effects of the vaccine (April–June 2021). Despite there being a significant amount of research investigating negative attitudes towards the COVID-19 vaccine, there is a lack of studies which explore the individual characteristics of those who are prone to receive the vaccine or who have already been vaccinated. Such knowledge can be useful in further understanding the mechanisms of individual acceptance of vaccination, in order to design and to foster health literacy about this issue more effectively. Therefore, we decided to control for the variability in attitude by excluding from our sample individuals with a negative attitude towards the vaccine.

More precisely, the aims of our study were (a) to classify into groups a sample of Italian people reporting a positive attitude towards the vaccine, on the bases of several individual differences, which are relevant for processing vaccine information, i.e., a tendency to believe in conspiracy theories, cognitive style, impulsivity, cognitive closure, and tolerance for ambiguity; and (b) to compare the change of vaccine risk-perception by investigating the two dimensions of probability and severity between the identified groups, after inconsistent communication of the side effects associated with the vaccine.

### 2.2. Procedure

Data were collected in Italy in the period following the reports of rare side-effects of the vaccine (30 April–8 June 2021). The survey was hosted and distributed using the Qualtrics online survey platform. A voluntary response sampling method was used: respondents were recruited via email invitations and advertisements on social media platforms. No incentive was given to complete the survey. 

Written informed consent was collected from all respondents. The study was approved by the Ethics Review Board of the Catholic University of the Sacred Heart of Milan.

### 2.3. Participants

Eighty respondents completed all of the sections of the survey (completion rate was 74.1%). One participant was excluded because they declared that they were opposed to the COVID-19 vaccine. A total sample of 79 participants, who reported a positive attitude towards the COVID-19 vaccine (i.e., they answered positively to the item “What is your attitude towards the COVID-19 vaccine?”), were considered in the analysis. Participants’ demographic information is reported in Table 1. 

### 2.4. Measures

*Socio-demographics*—A questionnaire investigating demographic variables, such as age, gender, educational level, region of residence, marital status, health state (“How are you feeling physically?”), employment status, news sources exposure (“How many sources of information do you consult daily to stay up-to-date on the current situation?”), and possible previous experience of COVID-19 infection was proposed to participants.

*Personal experience about COVID-19*—A questionnaire to investigate the personal experience about COVID-19 disease was designed. It was composed of five items to evaluate the experience of COVID-19 infection by participants and their relatives and the perceived probability of contracting it soon (i.e., “Were you sick with COVID-19?” (yes/no); “What was the severity of the symptoms?” (mild/moderate/severe but manageable at home/hospitalization); “How likely do you think a COVID-19 infection will be in the next few months?” (not at all/slightly probable/quite probable/very probable/at all); “Did one or more of your relatives or people close to you fall ill with COVID-19?” (yes/no); “What was the severity of the symptoms?” (mild/moderate/severe but manageable at home/hospitalization/death)).

*Attitude towards COVID-19 vaccine*—Two questions to delve into the attitude towards COVID-19 vaccine were designed (“Regardless of whether you have already been vaccinated against the COVID-19, what is your attitude towards the COVID-19 vaccine?” (extremely contrary/contrary/neither contrary nor favorable/favorable/entirely favorable); “If you think about your attitude towards the vaccine for COVID-19 before the start of the vaccination campaign, has it changed?” (much less favorable/slightly less favorable/unchanged/slightly more favorable/much more favorable)). Answers were rated on a 5-point Likert scale.

*Vaccine perceived risk*—An instrument to explore the vaccine’s perceived risk was designed. It was composed of four questions on a 5-point Likert scale (i.e., “I am very concerned about possible serious and/or irreversible side effects that the vaccine may have”; “Regardless of whether you have already been vaccinated against the COVID-19, what is your attitude towards the COVID-19 vaccine?”).

*Trust and conformism*—Three questions were developed to deepen the data regarding attitudes and social norms about the COVID-19 vaccine (i.e., “Most of the people close to me are against the vaccine” (R); “I trust health institutions; I trust scientific research”).

*Negative emotions*—Three negative high-arousal emotions towards vaccination (i.e., anger, anxiety, fear) were explored through a 5-point Likert scale (1 = not at all, 5 = very much). 

*Analytic thinking and impulsivity*—The cognitive reflection test (CRT, [56]) consisted of four questions (e.g., “A bat and a ball together cost 110 cents. The bat costs 100 cents more than the ball. How much does the ball cost?”—correct answer: 5 cents, intuitive answer: 10 cents). For each one, answers were coded as: correct and incorrect (distinguishing between intuitive/automatic and non-intuitive/non-automatic). The sum of the correct answers was considered as a measure of analytic thinking, whilst the sum of the intuitive/automatic answers as a measure of impulsivity.

*Impulsivity*—The S-UPPS-S impulsive behavior scale (S-UPPS-S, [57]) consisted of a self-reported questionnaire composed of 20 items on a 4-point Likert scale (1 = completely agree, 4 = completely disagree). It evaluated five aspects of impulsivity: positive urgency, negative urgency, lack of perseverance, lack of premeditation, and sensation seeking.

*Cognitive closure*—Need for cognitive closure (NCC, [58]) was a questionnaire composed of 42 items on a 6-point Likert scale (1 = strongly disagree, 6 = strongly agree). It investigated five dimensions: preference for order and structure, preference for predictability, decisiveness, discomfort with ambiguity, and closed-mindedness. 

*Attitude toward ambiguity*—The multidimensional attitude toward ambiguity scale (MAAS, [59]) was composed of 30 items on a 7-point Likert scale (1 = strongly disagree, 7 = strongly agree), which tap on three subscales: discomfort with ambiguity, moral absolutism/splitting, and need for complexity and novelty.

*Worry about COVID-19*—The COVID-19 worry scale (CWS, [60]) consists of seven items on a 4-point Likert scale, ranging from 1 (not at all) to 4 (very much). This self-reported measure assesses participants’ worry about the potential spread of COVID-19 and its consequences. Total scores could range from 7 to 28. A score of above 22 corresponds to individuals who are highly worried. The scale was translated into Italian from the original version according to a method of inter-judge agreement based on two native Italians’ independent translations and was subsequently translated by a third fellow expert in the original language to verify possible discrepancies with the original version.

*Conspiracy beliefs*—The generic conspiracist beliefs scale (GCB, [61]) is composed of a 15 item scale designed on a 5-point Likert scale (1 = definitely not true, 5 = definitely true) covering five dimensions: government malfeasance, malevolent global conspiracy, extraterrestrial cover-up, personal well-being, and control of information. The tool was translated into Italian through a method of inter-judge agreement based on two native Italians’ independent translations and subsequently translated by a third fellow expert in the original language.

### 2.5. Analytic Plan

First, reliability analyses (i.e., computing Cronbach’s α) were performed to test the internal consistency of all of the standardized measures. Then, the cognitive and behavioral characteristics which we considered relevant for processing vaccine information (i.e., tendency to believe in conspiracy theories, cognitive style, impulsivity, tolerance for ambiguity, cognitive closure) served as input variables for a cluster analysis. A two-phased clustering procedure was followed. More precisely, cluster validation was performed by comparing two different clustering techniques, namely, (a) a hierarchical cluster analysis, using the Euclidean distance measure and (b) a K-Means non-hierarchical cluster analysis, using the Hartigan–Wong algorithm. To further validate the optimal number of clusters for partitioning participants into distinct groups, we tested if the input measures differed significantly between the identified groups. When homogeneity of variance was violated, as measured by Levene’s test, Welch’s ANOVA was used. The variables which did not contribute significantly to the separation of the groups were then excluded in order to determine the most distinctive partitioning. Finally, concordance between the retained clusters from each method (i.e., hierarchical and K-Means analyses) was tested using Lin’s concordance correlation coefficient (CCC) [62].

Then, we compared the change of vaccine risk-perception between groups, considering two risk dimensions, namely: (a) probability (“Compared to before the start of the vaccination campaign, today I believe it is much more likely that serious and/or irreversible adverse effects may occur in me after the vaccine”); and (b) severity (“Compared to before the start of the vaccination campaign, as of today I am much more concerned about the adverse effects the vaccine may have”). The profile groups identified by the cluster analysis were added as a between-subject factor in the model. In addition, the role of the dependent variable was adjusted for two covariates which were theoretically related to risk-perception, namely: (1) vaccine trust and conformism (the mean of the “Most of the people close to me are against the vaccine reversed”, “I trust health institutions”, “I trust scientific research” items); and (2) high-intensity negative emotions (i.e., anger, anxiety, fear). To do so, a mixed-factorial ANCOVA was performed. Finally, to exclude the potential role of media exposure, we compared the number of consulted news sources between profile groups using one-way ANOVA. We calculated that a sample size of 79 was sufficient to detect a medium effect size (η^2^ = 0.06) in the mixed-factorial ANCOVA 3 × 2 with a power of 0.98 and alpha set at 0.05. 

## 3. Results

### 3.1. Reliability Analysis

All standardized measures had an acceptable internal consistency, as showed by the Cronbach’s α values computed for each scale (CRT: 0.72; GCB: 0.91; S-UPPS-S: 0.68; MAAS: 0.79; NCC: 0.86).

### 3.2. Cluster Analysis

The hierarchical cluster analysis identified three clusters. A K value of 3 was later specified in the K-Means analysis. The input variables which contributed significantly to the separation of the identified groups were retained and the K-Means analysis was retested to maximize the efficiency of the classification (Table 2). Results of the two cluster methods (i.e., hierarchical and K-Means analyses) showed high accordance (ρ_c_ = 0.86, 95% CI [0.79, 0.91]). 

The final solution identified three clusters of individuals with distinct cognitive/behavioral profiles (Figure 1). The larger cluster was labeled *Analytics* (*n* = 35) and included individuals who showed the highest relative levels of analytical reasoning and the lowest levels of decisiveness. We labeled as *Polarized* (*n* = 21) the cluster of individuals showing the lowest relative level of analytical reasoning (as showed by the highest frequency of impulsive incorrect responses and the lowest accuracy score in the CRT task) and the highest levels in all of the other dimensions, except for average sensation seeking and decisiveness. Finally, the third cluster, which we labeled as *Open* (*n* = 23), includes individuals with average analytical reasoning and the lowest relative scores in all dimensions, except for high levels of sensation seeking and decisiveness. 

### 3.3. Risk-Perception Change

A significant main effect of the risk dimension (probability vs. severity) was found (F(1,74) = 7.79; *p* = 0.007; η^2^ = 0.009); the side effects probability-perception changed less than the severity-perception (Figure 2). The main effect of the cluster profiles (i.e., analytics, polarized, open) was nonsignificant (F(2,74) = 0.88; *p* = 0.42). The interaction effect of risk dimension x cluster profiles was significant (F(2,74) = 3.32; *p* = 0.04; η^2^ = 0.008). The *post-hoc* analysis on the interaction between risk dimensions and cluster profiles revealed a significant difference between the risk-perception change between the probability and severity of side effects within the analytic group (t = 1.71; p_tukey_ = 0.004). In this profile group, the severity change was significantly larger than the probability change. Such a difference was nonsignificant for the other two cluster profiles (i.e., polarized, open). Finally, both covariates are significant predictors of risk-perception change (vaccine trust and conformism: F(1,74) = 5.97; *p* = 0.02; η^2^ = 0.022; high-intensity negative emotion: F(1,74) = 15.78; *p* < 0.001; η^2^ = 0.06), whereas news source exposure did not differ between profile groups (F(2,76) = 1.23; *p* = 0.30).

## 4. Discussion

The present article contributes to a deeper understanding of who pro-vaccine people are, by identifying three different personality and cognitive profiles within a group of Italian people reporting a positive attitude toward COVID-19 vaccine. Our findings suggest that, depending on individual differences in terms of a tendency to believe in conspiracy theories, cognitive style, impulsivity, tolerance for ambiguity, and cognitive closure, they can be grouped into analytic, polarized, and open profile clusters.

Specifically, the analytic profile cluster tends to be involved in logical and deliberative reasoning, being skeptical about conspiratorial assumptions, presenting the lowest decisiveness and overcoming intuitive or impulsive solutions. This is in line with studies investigating analytical thinking and personal beliefs (i.e., [33,63,64]). Polarized individuals seem to share some features with the so-called “authoritarian” personality [65], which is typical of people who are intolerant to ambiguity and tend to think in black-and-white, or think dichotomically. In line with more recent literature, this lack of cognitive flexibility turns out to be associated also to beliefs in conspiracy theories [33]. Moreover, the polarized profile cluster presents the lowest level of analytical reasoning and the highest levels in impulsivity and in the other investigated individual differences, except for the dimensions of sensation seeking and decisiveness. Coherently, it is assumed that high levels of NCC can increase the urge to accept the first available solution and go ahead on it, in order to reduce the ambiguity of a given situation, which therefore is reflected in being impulsive [40].

Finally, in line with the literature, we found an open profile cluster which identifies people who are receptive to new and unfamiliar information and ideas, counterevidence, and ambiguity [66,67,68]. These people, in our sample, are also characterized by both high levels of decisiveness and low levels of all other NCC subscales. Accordingly, it provides support to the existence of two distinct and divergent dimensions within the NCC construct, a “seizing” process (characterized by decisiveness and a desire to obtain a fast and aspecific solution) and a “freezing” process (denoted by anchoring to specific solutions to respond to a given event, through the use of simple structures) [69,70]. Thus, the decisiveness dimension leads directly to a conclusion, even if it is an erroneous one.

How did these three groups change their perception of risk after inconsistent news of side effects associated with the vaccine occurred during the first months of the vaccination campaign? In line with the literature on subjective risk-perception highlighting that sensational and low frequency risks tend to be overestimated [53,71,72,73], we found that the occurrence of serious, albeit infrequent, side effects have impacted on the risk-perception of the vaccine itself. However, our results suggest that, even after excluding the potential role of media exposure, the risk-perception analyzed according to the two dimensions of probability and severity of the consequences [50,51] changes over the course of the vaccination campaign, following different trends within analytical, polarized, and open people. Analytical people are those who, in a functional way, have increased their perception of the severity regarding the COVID-19 vaccine’s side effects (according to media communication regarding the isolated deaths after a vaccine dose), but leaving the perception of probability almost unchanged (probably based on the incidence of cases in which serious side effects occurred, which have remained fairly similar over time). Therefore, they were able to keep severity and probability separated and to modify the risk severity estimation accordingly. A possible explanation can be that the ability to engage in an analytical, critical evaluation of the situation, which characterizes this profile cluster, leads to processing the available information in a more functional and in-depth way. We can assume that such people were not biased by the sensational media communication of isolated adverse events regarding the occurrence of severe vaccine’s side effects, probably not being influenced, at least in part, by possible heuristics (i.e., availability heuristic [55]). 

Instead, both polarized and open people—in their estimation of risk associated with vaccines—did not discriminate between probability and severity, thus indicating a risk-perception that remains unchanged during the succession of events. The similar trend in risk estimation—as shown in both polarized and open people—should probably be attributed to different causes/factors, that may arise from a tendency to process information in a less in-depth manner than the analytic group. In detail, polarized people, who present the features described above, are probably more prone to jump to conclusions. In fact, the literature suggests that individuals who are prone to jumping to conclusions appear to be lower in cognitive ability and tend to rely more on intuitive thinking, being more likely to respond impulsively to the CRT [74], and not analyzing the information in depth [74,75].

However, open people may prefer novelty in order to be engaged in innovative behaviors and health prevention. Despite their curiosity, openness, and their high tolerance of ambiguity, they tend to move quickly to a conclusion, even if it is misleading or erroneous. Thus, according to the literature, openness can lead to heuristic thinking, resulting in the use of less logical strategies [76]. 

We can assume that this tendency to think impulsively and to overshadow a rational evaluation of available data results in an oversimplification of reality, which is typical of thinking in a dichotomic and rigid mode. This may result in people being more exposed to cognitive heuristics. 

Considering the differences between the open and polarized profile clusters, the first one, despite the curiosity, openness and high tolerance of ambiguity which characterize it, tends to move quickly to a conclusion, even a misleading and erroneous one.

This difference among the three groups remains even after controlling for possible variables of influence, specifically negative emotions and trust and conformism, which can affect the perception of risk (i.e., [8,77,78,79,80]).

Accordingly, analytical reasoning plays a crucial role in the ability and the willingness of individuals to engage in processes of complex thinking, like those required in contexts that are constantly changing and characterized by information ambiguity.

### Limitations

The present study has some limitations. 

Firstly, the cross-sectional design does not allow for conclusions regarding causal relations and it provides only an indirect measure about possible changes over time. Secondly, participants completed the study online, so they might be subject to confounding variables (i.e., concerning the reliability of participants in their home environment) or social desirability bias. Moreover, the sample of the present research is not representative of the Italian population, due to the mean age of participants (30.2 years). Further studies are needed to investigate the risk-perception in older age groups as well. Besides, further research should deepen the dimensions of perceived risk associated with the COVID-19 vaccine and personality measures which better discriminate between individual profiles. Additionally, it could be interesting to propose the present study to participants who exhibit a negative attitude toward the COVID-19 vaccine, to explore possible individual profiles and compare them with the others described previously. 

## 5. Conclusions

Although many studies have investigated negative attitudes towards the COVID-19 vaccine, there is a lack of studies that explore the individual characteristics of those who are prone to receive the vaccine or who have already been vaccinated. The present study explores the two dimensions of risk-perception that so far have been poorly investigated especially in relation to health promotion behaviors [49] in people who are pro COVID-19 vaccine. Accurate risk-perception can help citizens to make well-informed decisions towards risks and mobilize actions for prevention [53]. Specifically, to increase the likelihood that people who have made their first/second dose(s) which will complete the vaccination series, it is crucial to understand how risk-perception changes in people who are pro-vaccine in a situation characterized by ambiguity and inconsistent information. In this way, the present results highlight the importance of the communication of events. Specifically, conflicting or sensationalized information in uncertain contexts can lead some people, characterized by specific individual differences, to process available data in a non-functional way. This can result in possible negative consequences for them (such as poor health choices regarding preventive behaviors or hesitancy toward the vaccination or the completion of the vaccination cycle) and, more in general, for public health (resulting in common practices and attitudes that undermine collective safety [3]).

The present findings suggest some practical implications, both in the short-medium and in the long term. For the prior one, it is desirable that healthcare providers offer exhaustive and clear information—through multiple media [81]—to citizens to enable them to consciously make their health choices, with the awareness that people present different cognitive resources to process available data. In fact, difficult or biased ways of reporting data may lead some people to adopt behaviors based on their representation of risk, rather than on the objective risk [82]. Thus, communication about the pandemic and its related issues (included vaccines) plays a key role in shaping judgments about the risk associated to an event [83]. In the long term, these results could be useful at a formative-educational level, with the aim of designing tailored interventions [81] to help enhance cognitive characteristics which are important for information processing and decision making, such as analytical thinking, especially in individuals who are more vulnerable. Finally, the present study can add knowledge to further understand the mechanisms of individual acceptance of vaccination and to design and foster health literacy about this issue more effectively. 

## Figures and Tables

**Figure 1 ijerph-19-01189-f001:**
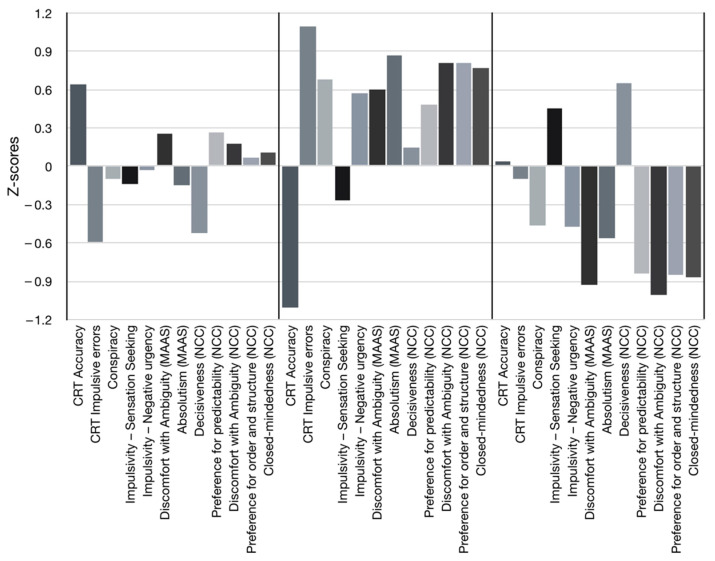
Three-cluster solution profiles, based on the following input measures: (1) CRT accuracy; (2) CRT impulsive errors; (3) conspiracy; (4) impulsivity—sensation seeking; (5) impulsivity—negative urgency; (6) discomfort with ambiguity (MAAS); (7) absolutism (MAAS); (8) decisiveness (NCC); (9) preference for predictability (NCC); (10) discomfort with ambiguity (NCC); (11) preference for order and structure (NCC); (12) closed-mindedness (NCC).

**Figure 2 ijerph-19-01189-f002:**
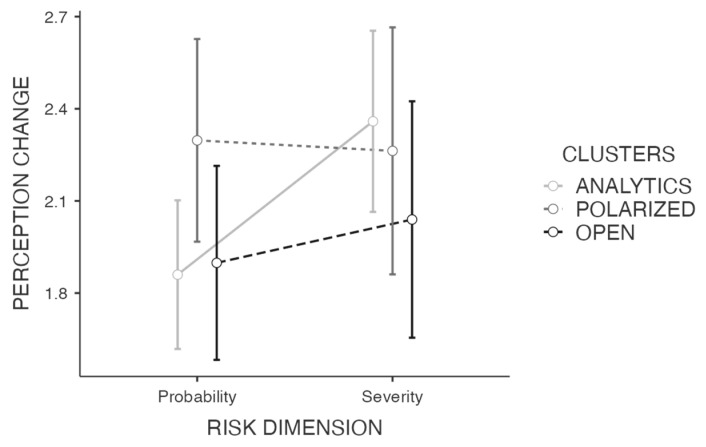
Comparison of the risk-perception (i.e., probability and severity) change between cluster profiles (i.e., analytics, polarized, open), with vaccine trust and conformism and high-intensity negative emotions as covariates.

**Table 1 ijerph-19-01189-t001:** Participants’ characteristics.

	*n*	%
Age (M, SD)	30.2	13.0
Gender		
Male	29	36.7
Female	50	63.3
Educational level		
Middle school	1	1.2
High school	28	35.4
Bachelor’s degree	21	26.6
Graduate/Master	16	20.3
MD/PhD	10	12.7
Other	3	3.8
Region		
Northern Italy	60	75.9
Center Italy	12	15.19
Southern Italy	7	8.86
Marital status		
Single	56	70.9
Married/partnered	21	26.6
Divorced/widowed	2	2.5
Health state		
Quite good	24	30.4
Good	34	43.0
Very good	21	26.6
Employment status		
Students	41	51.9
Employed	33	41.8
Retired	5	6.3
News sources quantity (M, SD)	2.82	1.28
Previous COVID-19 infection	10	12.7

**Table 2 ijerph-19-01189-t002:** Cognitive and behavioral dimensions z-scores for the final 3-cluster solution, and their comparison between the classified groups.

	Analytics	Polarized	Open	Comparison (F, *p*)
CRT Accuracy	0.64	−1.11	0.04	50.36, 0.001
CRT Impulsive Errors	−0.59	1.10	−0.10	33.33, 0.001
Conspiracy	−0.10	0.68	−0.46	8.91, 0.001
Impulsivity—Sensation Seeking	−0.14	−0.27	0.45	3.56, 0.037
Impulsivity—Negative Urgency	−0.03	0.57	−0.47	6.35, 0.004
Discomfort with Ambiguity (MAAS)	0.26	0.60	−0.93	24.46, 0.001
Absolutism (MAAS)	−0.15	0.87	−0.56	14.59, 0.001
Decisiveness (NCC)	−0.52	0.15	0.65	13.45, 0.001
Preference for Predictability (NCC)	0.27	0.48	−0.84	19.78, 0.001
Discomfort with Ambiguity (NCC)	0.18	0.81	−1.01	26.71, 0.001
Preference for Order and Structure (NCC)	0.07	0.81	−0.85	19.81, 0.001
Closed-mindedness (NCC)	0.11	0.77	−0.87	17.62, 0.001

CRT: cognitive reflection test; MAAS: multidimensional attitude toward ambiguity scale; NCC: need for cognitive closure.

## Data Availability

Data presented in this study are available on request from the corresponding author. Data are not publicly available due to participants did not accept to share their data with third people besides the research team.

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
