# Peer review of "Risk-Perception Change Associated with COVID-19 Vaccine’s Side Effects: The Role of Individual Differences"

_ijerph, 2022, doi:10.3390/ijerph19031189_

Round 1

Reviewer 1 Report

This article assesses the role of individual differences in risk-perception change associated with COVID-19 vaccine’s side effects.  Authors assert a need for this study because while COVID-19 vaccine is a crucial requirement in fighting the pandemic, conspiracy theories and negative attitudes towards the vaccine alters the population’s perceived risk of contacting COVID-19.  Authors also argue that vaccine hesitancy has been one of the greatest threats to public health at the global level.  Therefore, this study uses an online survey to measure the effect of individual differences in the risk-perception related to the COVID-19 vaccine’s side effects in a context depicted by information ambiguity.

Feedback is as follows:

  1. The aim of the study is presented in the Materials and Methods section but should be introduced much earlier in the paper. Authors should consider stating the study aim at the end of the Introduction section.
  2. Lines 39-44, if the numbers presented are for Italy, the authors should indicate so. This could be done by rephrasing line 39 to “In detail, Italy’s eighth Covid-19 Vaccine Pharmacovigilance Report described…”.
  3. Line 54 – In “on the 15th March 2021 some European country”, it should be ‘countries’ (plural).
  4. For the Conspiracy Theories section (lines 70-90), have any of the cited studies been conducted in Italy? If not, the authors further can emphasize this as a rationale for their study and how their study adds to the existing literature.
  5. Line 156 – Check wording in “Despite there is a lot of research which investigated the negative attitude towards the Covid-19 vaccine”.
  6. For the study procedures, what sampling technique was used for recruiting participants? Was it a snowball sampling method?
  7. Were participants offered an incentive (e.g., payment) to complete the survey?
  8. Line 179 – What was the response rate for survey completion?
  9. Line 184 – In Table 1, ‘COVID-19 infection’ is not characterized like the rest of the variables. Later in the paper (line 188, authors describe “previous experience of Covid-19 infection”.  Perhaps this can be a label for the variable in Table 1 based on COVID infection status (e.g., yes, no, unsure) among the participants.
  10. Line 186 – Was other sociodemographic information collected such as employment status and income level?
  11. The Results presented are intricate and at times hard to follow. Although, authors do a sound job in clearly explaining the results in the Discussion section.
  12. Line 429 – What are the “possible negative consequences”? Authors should expand.
  13. Overall, the Conclusions section effectively summarizes the paper.

In summary, this is an insightful, pertinent, and unique study on a very important topic.  Attending to some clarifying questions, including about the materials and methods, may help to improve the overall paper.

Author Response

This article assesses the role of individual differences in risk-perception change associated with COVID-19 vaccine’s side effects.  Authors assert a need for this study because while COVID-19 vaccine is a crucial requirement in fighting the pandemic, conspiracy theories and negative attitudes towards the vaccine alters the population’s perceived risk of contacting COVID-19.  Authors also argue that vaccine hesitancy has been one of the greatest threats to public health at the global level.  Therefore, this study uses an online survey to measure the effect of individual differences in the risk-perception related to the COVID-19 vaccine’s side effects in a context depicted by information ambiguity.

Feedback is as follows:

  1. The aim of the study is presented in the Materials and Methods section but should be introduced much earlier in the paper. Authors should consider stating the study aim at the end of the Introduction section.

RE: We followed the reviewer’s suggestion and we mentioned the study aim at the end of the introduction section.

  1. Lines 39-44, if the numbers presented are for Italy, the authors should indicate so. This could be done by rephrasing line 39 to “In detail, Italy’s eighth Covid-19 Vaccine Pharmacovigilance Report described…”.

RE: Thanks for this remark. We reported the sentence as the reviewer suggested.

  1. Line 54 – In “on the 15th March 2021 some European country”, it should be ‘countries’ (plural).

RE: Thanks for identifying this mistake, which was emended in the revised version of the manuscript.

  1. For the Conspiracy Theories section (lines 70-90), have any of the cited studies been conducted in Italy? If not, the authors further can emphasize this as a rationale for their study and how their study adds to the existing literature.

RE: Our study is not the first one on this topic. Hence, we cited other studies that have been conducted in Italy, whose results are coherent with the others previously cited.

  1. Line 156 – Check wording in “Despite there is a lot of research which investigated the negative attitude towards the Covid-19 vaccine”.

RE: We have reworded the sentence.

  1. For the study procedures, what sampling technique was used for recruiting participants? Was it a snowball sampling method?

RE: We used a voluntary response sampling method. This is now specified in the Procedure section.

  1. Were participants offered an incentive (e.g., payment) to complete the survey?

RE: No incentive was given to the participants. We reported this piece of information in the revised version of the paper.

  1. Line 179 – What was the response rate for survey completion?

RE: Of 108 respondents who opened the survey, 80 participants completed all its sections and were included in the study. Completion rate was approximately 74%. This information has been now added to the Participant section.

  1. Line 184 – In Table 1, ‘COVID-19 infection’ is not characterized like the rest of the variables. Later in the paper (line 188, authors describe “previous experience of Covid-19 infection”.  Perhaps this can be a label for the variable in Table 1 based on COVID infection status (e.g., yes, no, unsure) among the participants.

RE: Thank you for your suggestion. Indeed, the ‘COVID-19 infection’ variable referred to previous experience of Covid-19 infection. We have now changed the label of the variable in Table 1 to make it clearer.

  1. Line 186 – Was other sociodemographic information collected such as employment status and income level?

RE: Employment status (student/employed/retired) was collected. This information has been now added to the participants’ characteristics. We did not measure participants’ income level in order to prevent that participants were induced to stop filling the survey because they had to answer a question about a sensible issue.

  1. The Results presented are intricate and at times hard to follow. Although, authors do a sound job in clearly explaining the results in the Discussion section.

RE: We tried to change the wording of some of the sentences in the Results section, in the attempt to make them clearer.

  1. Line 429 – What are the “possible negative consequences”? Authors should expand.

RE: We have specified the possible negative consequences in the Conclusions section.

  1. Overall, the Conclusions section effectively summarizes the paper.

In summary, this is an insightful, pertinent, and unique study on a very important topic.  Attending to some clarifying questions, including about the materials and methods, may help to improve the overall paper.

RE: Authors thank the Reviewer for comments and considerations made above.

Reviewer 2 Report

The manuscript titled “Risk-perception Change Associated with COVID-19 Vaccine’s Side Effects: The Role of Individual Differences” proposed an investigation on the individual differences affecting the change of risk perception associated with Covid-19 vaccine’s side effects, in a context characterized by information ambiguity. Seventy-nine Italian pro-vaccine participants were recruited throughout an online survey investigating several domains and were then clustered into 3 cognitive/behavioral style groups (Analytic, Open, Polarized). The change of risk perception about vaccines after inconsistent communications of vaccine’s side effects was compared between groups, together with some covariates. Results showed that Analytic participants changed the perception of the severity of side effects, but they did not change their estimation of probability of vaccine side effects, whilst “Open” and “Polarized” participants did not change neither of the probability nor of the severity of side effects. Authors discussed their results in light of previous literature highlighting strengths and limitations of their work, suggesting also hints for further research.

I carefully read the manuscript, and I think it may be of interest for the readers of International Journal of Environmental Research and Public Health. The manuscript is very well-written and properly addresses the interesting issue of the factors associated with vaccine’s side effects risk perception. The topic is really relevant nowadays, and more knowledge coming from well-conducted primary studies is needed. I also appreciated the construction of a well-structured survey in order to provide a complete representation of the phenomenon. I found that the introduction section and the methodology employed are clear and detailed, as well as the explanations provided in the discussion section. I only have few minor remarks:

Introduction section

Line 42-43: The use of percentage of 0.72% per 100,000 doses might be confusing. According to that percentage, a number of 84,010,605*0.72% is equal to 604,876 deaths from the beginning of the vaccination campaign. Please, can you provide the absolute frequency of deaths per 100,000 doses?

Line 67: please remove the link to the website www.reverso.net. I found such links also in other lines of the manuscript.

Materials and Methods section

Measures subsection

Line 187 and Table 1: Please declare which kind of item was used to appraise the health state.

My suggestion is also to calculate and report a reliability index (e.g., Cronbach’s alpha) for the various questionnaires employed, when applicable.

Discussion section

Line 353: you ask “how did these three groups change their perception of risk after inconsistent news of side effects associated with the vaccine? I agree that this is the main question of the manuscript, and that’s a very relevant question. I have some doubts on the following points: 1) Might all the participants included agree with you that there were “inconsistent news” on the first vaccination weeks? 2) Did you take some control measure regarding the exposure, in terms of both quality and quantity, to media and news about vaccination? How can you be confident that all the participants were exposed to the same sources of information and that such exposure was not another key factor modulating risk perception?

I think this is a really relevant topic, and for your convenience please find attached references of three recent papers which can corroborate your work on Covid-19 as well as on vaccines’ risk perception:

1) Iachini, T., Frassinetti, F., Ruotolo, F., Sbordone, F. L., Ferrara, A., Arioli, M., . . . Ruggiero, G. (2021). Social distance during the covid-19 pandemic reflects perceived rather than actual risk. International Journal of Environmental Research and Public Health, 18(11) doi:10.3390/ijerph18115504

2) de Sousa, Á. F. L., Teixeira, J. R. B., Lua, I., de Oliveira Souza, F., Ferreira, A. J. F., Schneider, G., ... & Fronteira, I. (2021). Determinants of COVID-19 Vaccine Hesitancy in Portuguese-Speaking Countries: A Structural Equations Modeling Approach. Vaccines, 9(10), 1167. doi:10.3390/vaccines9101167

3) Konstantinou, P., Georgiou, K., Kumar, N., Kyprianidou, M., Nicolaides, C., Karekla, M., & Kassianos, A. P. (2021). Transmission of Vaccination Attitudes and Uptake Based on Social Contagion Theory: A Scoping Review. Vaccines, 9(6), 607. doi:10.3390/vaccines9060607

Author Response

Comments and Suggestions for Authors

The manuscript titled “Risk-perception Change Associated with COVID-19 Vaccine’s Side Effects: The Role of Individual Differences” proposed an investigation on the individual differences affecting the change of risk perception associated with Covid-19 vaccine’s side effects, in a context characterized by information ambiguity. Seventy-nine Italian pro-vaccine participants were recruited throughout an online survey investigating several domains and were then clustered into 3 cognitive/behavioral style groups (Analytic, Open, Polarized). The change of risk perception about vaccines after inconsistent communications of vaccine’s side effects was compared between groups, together with some covariates. Results showed that Analytic participants changed the perception of the severity of side effects, but they did not change their estimation of probability of vaccine side effects, whilst “Open” and “Polarized” participants did not change neither of the probability nor of the severity of side effects. Authors discussed their results in light of previous literature highlighting strengths and limitations of their work, suggesting also hints for further research.

I carefully read the manuscript, and I think it may be of interest for the readers of International Journal of Environmental Research and Public Health. The manuscript is very well-written and properly addresses the interesting issue of the factors associated with vaccine’s side effects risk perception. The topic is really relevant nowadays, and more knowledge coming from well-conducted primary studies is needed. I also appreciated the construction of a well-structured survey in order to provide a complete representation of the phenomenon. I found that the introduction section and the methodology employed are clear and detailed, as well as the explanations provided in the discussion section. I only have few minor remarks:

Introduction section

Line 42-43: The use of percentage of 0.72% per 100,000 doses might be confusing. According to that percentage, a number of 84,010,605*0.72% is equal to 604,876 deaths from the beginning of the vaccination campaign. Please, can you provide the absolute frequency of deaths per 100,000 doses?

RE: Thank you. We have provided the absolute frequency of deaths.

Line 67: please remove the link to the website www.reverso.net. I found such links also in other lines of the manuscript.

RE: Sorry, but unfortunately we did not find the link in the text. Maybe it is no longer visible.

Materials and Methods section

Measures subsection

Line 187 and Table 1: Please declare which kind of item was used to appraise the health state.

My suggestion is also to calculate and report a reliability index (e.g., Cronbach’s alpha) for the various questionnaires employed, when applicable.

RE: We added information about the health status measure in the Measures section. Furthermore, Cronbach’s alphas of the standardized questionnaires are now reported in the Results section.

Discussion section

Line 353: you ask “how did these three groups change their perception of risk after inconsistent news of side effects associated with the vaccine? I agree that this is the main question of the manuscript, and that’s a very relevant question. I have some doubts on the following points: 1) Might all the participants included agree with you that there were “inconsistent news” on the first vaccination weeks? 2) Did you take some control measure regarding the exposure, in terms of both quality and quantity, to media and news about vaccination? How can you be confident that all the participants were exposed to the same sources of information and that such exposure was not another key factor modulating risk perception?

RE:

1) As regards inconsistent news, we referred to the communication messages that had been widespread in Italy between January and June (mentioned in the Introduction) about what has been recommended and adjusted by AIFA. Since communication was changing rapidly, we did not include a specific question regarding this point.

2) As regards media exposure, we collected a measure of how many sources of  information had been used by participants. This has now been added in the section about Participants’ characteristics. Furthermore, we did a comparison of the sources quantity between clusters to support the notion that the three groups were exposed to a similar amount of news and information about the Covid.

I think this is a really relevant topic, and for your convenience please find attached references of three recent papers which can corroborate your work on Covid-19 as well as on vaccines’ risk perception:

1) Iachini, T., Frassinetti, F., Ruotolo, F., Sbordone, F. L., Ferrara, A., Arioli, M., . . . Ruggiero, G. (2021). Social distance during the covid-19 pandemic reflects perceived rather than actual risk. International Journal of Environmental Research and Public Health, 18(11) doi:10.3390/ijerph18115504

2) de Sousa, Á. F. L., Teixeira, J. R. B., Lua, I., de Oliveira Souza, F., Ferreira, A. J. F., Schneider, G., ... & Fronteira, I. (2021). Determinants of COVID-19 Vaccine Hesitancy in Portuguese-Speaking Countries: A Structural Equations Modeling Approach. Vaccines, 9(10), 1167. doi:10.3390/vaccines9101167

3) Konstantinou, P., Georgiou, K., Kumar, N., Kyprianidou, M., Nicolaides, C., Karekla, M., & Kassianos, A. P. (2021). Transmission of Vaccination Attitudes and Uptake Based on Social Contagion Theory: A Scoping Review. Vaccines, 9(6), 607. doi:10.3390/vaccines9060607

RE: We have added these relevant papers.

Authors thank the Reviewer for comments and considerations made above.

Round 2

Reviewer 1 Report

The authors have responded well to the suggested feedback and the paper is clearer and improved.  One minor suggestion would be to still move the aim of the paper to earlier in the paper, possibly to the end of the third paragraph in the Introduction.  This can help establish the purpose of the paper earlier in the manuscript.

Author Response

The authors have responded well to the suggested feedback and the paper is clearer and improved.  One minor suggestion would be to still move the aim of the paper to earlier in the paper, possibly to the end of the third paragraph in the Introduction.  This can help establish the purpose of the paper earlier in the manuscript.

RE: Thank you for the feedback and the suggestion; We have moved the aim to the end of the third paragraph in the Introduction.
